# Collagen imaging reveals synergistic effects of sutures and host-donor misalignment on topographical irregularities in penetrating keratoplasty

**Himanshu Gururani[1], Sai Naga Sri Harsha Chittajallu[1], Minal Doulatramani[3], Viswanath Chinthapenta[1], Sayan Basu[2,3], Ramji M.[4]** *

1 Department of Mechanical and Aerospace Engineering, Micro-Mechanics Lab, Indian Institute of Technology Hyderabad, Sangareddy, Telangana, India, 2 Professor Brien Holden Eye Research Centre, L V Prasad Eye Institute, Hyderabad, Telangana, India, 3 The Cornea Institute, LV Prasad Eye Institute, Hyderabad, Telangana, India, 4 Department of Mechanical and Aerospace Engineering, Engineering Optics Lab, Indian Institute of Technology Hyderabad, Sangareddy, Telangana, India

* ramji_mano@mae.iith.ac.in

**Data Availability Statement:** All relevant data are within the manuscript and its Supporting Information files.

## Abstract

### Purpose

Mitigating unwanted refractive errors is crucial for surgeons to ensure quality vision after penetrating keratoplasty (PK). The primary objective of the present study is to highlight the importance of microstructural matching of the host and the donor cornea during PK on the distribution of the corneal tissue while suturing.

### Methods

For this purpose, the present study undertakes an *in-vitro* PK model to analyse the effect of suturing and host-donor misalignment on corneal birefringence. Five groups of experiments were performed using five corneoscleral buttons. In each group, N = 16 data points (corresponding to 16 simple interrupted sutures) were assessed before and after PK with five degrees of misalignments, 0°, 30°, 45°, 60° and 90° to detect the variations in corneal birefringence post-PK. The technique of digital photoelasticity is utilized to capture the corneal birefringence experimentally.

### Results

The local and global features of corneal birefringence provided interesting insights into the nuances of corneal birefringence in PK. Statistical analysis was performed to study the effects of suturing on the birefringence around the suture bites. It was observed that the interaction of the suture tension and structural misalignment between the host and the donor cornea influences the corneal birefringence in PK. **Conclusions**

   The zero-degree structural misalignment of the host and the donor tissue is preferable to minimize the topographical irregularities and related astigmatism post-PK. The findings of

**Funding:** This study was funded by the intramural funds of LV Prasad Eye Institute (LVPEI), Hyderabad; and partially sup- ported under Prime Minister Doctoral Research Fellowship grant (to author HG) by the Department of Science and Technology, India, Sci- ence, and Engineering Research Board, India, Confederation of Indian Industries, India, the Government of India & LVPEI.

**Competing interests:** NO authors have competing interests

the present study envisage an additional step of structurally aligning the donor tissue with the host before suturing to minimize topographical irregularities in PK.

## Introduction

The quality of vision is governed by corneal topography, which is a consequence of the equilibrium between the physiological forces acting on the cornea, its mechanical strength, and related pathology. In any surgical intervention involving alteration in the corneal curvature, there exists a shift in the physiological parameters such as IOP, geometry, and material properties of the cornea for a short time. After the surgery, the primary requirement is that the cornea acquires its original configuration to attain normal visual acuity. To achieve this objective, the clinicians meticulously consider every possible aspect of the clinical routine that can affect the corneal topography [1]. Typically, the clinical routines are classified as pre-operative evaluation, intra-operative management, and post-op care.

Penetrating Keratoplasty (PK) is a widely adopted surgical procedure involving trephining the host cornea's diseased part and replacing it with healthy donor tissue. It is performed for various ocular conditions such as corneal scarring, corneal dystrophies, keratitis, keratoconus, and failed/rejected grafts. Notably, around 20% of the patients undergoing PK develop astigmatism (greater than five dioptres) post-op, leading to severely compromised visual acuity [2, 3]. The factors contributing toward post-penetrating Keratoplasty astigmatism (PPKA) [4, 5] can be classified into pre-operative, intra-operative, and post-operative. Some of the intraoperative techniques that mitigate the PPKA are adjustment of suture tension, suture length, torque, and different suturing techniques [6, 7]. Further, the techniques such as sequential suture removal (SSR) [8, 9], insertion of intra-stromal ring (ICRS) [10], intraocular lens (IOLs) [11, 12], Photorefractive Keratectomy [13, 14], laser in-situ keratomileusis (LASIK) [15, 16] and repeat keratoplasty [17] are used post-operatively to manage PPKA.

The cornea comprises a dense network of collagen fibrils dispersed in a hydrated matrix. The collagen fibrils are arranged in a specific manner to provide optical transparency and mechanical strength to the corneal tissue [18, 19]. Various biomechanical studies have shown an anisotropic behaviour of the cornea due to its collagen fibre architecture [19–22]. The current surgical practice does not account for microstructural matching, leading to the random orientation of the donor cornea with respect to the host eye [23], which could be a risk factor contributing to PPKA. Richhariya et al. [24] showcased the multifaceted character of corneal astigmatism after PK through a simulation-based study using the finite element method. As per their study, astigmatism generated after PK is a cumulative effect of suture tension and orientation disparity between the donor and the host tissue.

Presently, the techniques that manage the PPKAs are primarily based on evaluating the corneal curvature [25, 26]. These techniques involve assessing the factors such as suture length, suture depth, suture tension, suture radiality, and its distance from the optical centre of the cornea. While performing PK, it is assured that all the above factors are nearly the same for all the sutures [15, 27]. The host-donor structural misalignment is an unaddressed factor that may have implications on PPKA. Therefore, the focus of the current study is to showcase the importance of microstructure matching of the host and the donor cornea to avoid the occurrence of PPKA associated with the anisotropic behaviour of the cornea.

Polarimetry is a non-invasive technique that gives information on the collagen fibre architecture of the cornea in the form of interference patterns [28–34]. It measures the optical

property exhibited by the cornea, known as birefringence. Previously, the authors have shown that the birefringence of the cornea is the cumulative effect of microstructure, stress distribution, and curvature [32]. The authors propose that the microstructural mismatch resulting from the host-donor misalignment during the corneal transplantation procedure may contribute to the topographic irregularities leading to PPKA. The primary objective of the present study is to highlight the importance of intraoperative microstructural matching of the host and the donor cornea during PK on distribution of the corneal tissue while suturing. For this purpose, the current study undertakes an in-vitro PK model to highlight the importance of host-donor misalignment by measuring the birefringence properties of the human cadaver corneoscleral buttons using the digital photoelasticity technique.

The current article is composed as follows: Section 'Introduction' lays out the background on the management strategies used to minimize PPKA and establishes the need to investigate the role of microstructural mismatch between the host and the donor cornea during PK. Sections 'Experimental Set-up', 'Specimen Preparation', 'Data Extraction' & 'Data Analysis' describes the experimental protocol to investigate the influence of suturing and host-donor misalignment on corneal birefringence. In this section, the technique of digital photoelasticity that measures birefringence is briefly explained. Section 'Results' outlines the results on the corneal birefringence obtained for various cases of misalignments. Section 'Discussion', which discusses the effect of suturing and misalignment on corneal birefringence and describes their role in creating topographic irregularities and concludes the present study. The authors present the hypothesis that the corneal birefringence will change with host-donor misalignment compared to the perfectly aligned case. The rationale behind the proposed hypothesis is as follows: when the host and donor corneas are sutured together in misaligned scenarios, the native structural anisotropy is disturbed. And the reaction forces generated by the sutures placed on the undesirable locations will redistribute the stresses around the suture bites, resulting in birefringence and topographic change. Therefore, the null hypothesis is defined as follows: The host-donor misalignment will not change the measured birefringence around the suture bites.

## Methods

The following section elucidates the methods used in the present study to test the proposed hypothesis. The experimental technique of digital photoelasticity to obtain birefringence data is briefly introduced in 'Experimental Set-up'. The details of the specimen preparation, data extraction, and data analysis are presented in Sections 'Specimen Preparation' to 'Data Analysis', respectively.

### Experimental set-up

The technique of digital photoelasticity provides full-field information on the birefringence of the specimen under study. The experimental set-up consists of a polariscope in the plane and circular configuration. **Fig 1** gives the schematic of a generic circular polariscope. The birefringence data is obtained by digital processing of multiple images acquired by the phase-shifting technique [35]. The ten-step phase-shifting technique [36] was used to estimate the isoclinic and isochromatic phase maps of the cornea.

The raw RGB (red-blue-green)-indexed images of each cornea was imported into MATLAB and then converted to double precision images. These images were then processed using an arctan function in MATLAB to extract the isoclinic phase map of the cornea. The arctan function gives the isoclinic values wrapped in the interval $\left[-\frac{\pi}{4}, \frac{\pi}{4}\right]$. Further, the adaptive quality guided phase unwrapping algorithm [36] was written in MATLAB to unwrap the isoclinic phase map. Later, the wrapped isochromatic phase map was calculated using an arctan

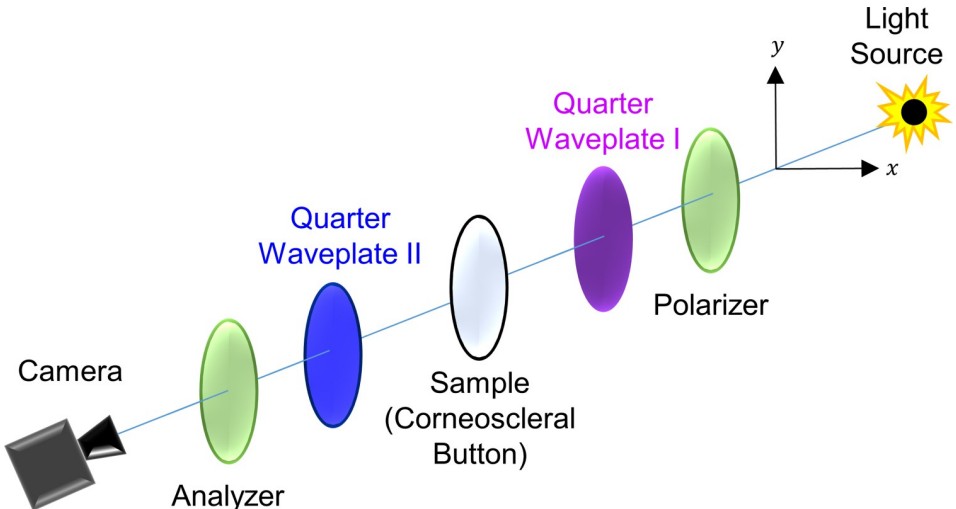

**Fig 1. A generic transmission polariscope in the circular configuration used for the birefringence imaging of the human cornea.**

function. A similar quality guided phase unwrapping algorithm was written in MATLAB to unwrap the isochromatic phase map of the cornea. A complete description of the experimental setup and methodology to extract birefringence data from the raw photoelastic images is given in the previous work conducted by the authors [32]. The raw images photoelastic images of the all the corneas is provides as a suppoting information in S1 File.

## Specimen preparation

Twelve Corneoscleral buttons were collected from the Ramayamma international eye bank of LV Prasad Eye Institute, Hyderabad, India. The buttons were identified as suitable for the corneal transplantation purpose by the eye bank based on their optical quality and therefore were provided for the research purpose. However, it is important to note that the corneal buttons are evaluated for their optical quality, endothelial cell density, and thickness to identify them as the transplantable grade. In the present study, the buttons used for birefringence measurement are of optimum optical quality (transparency) so that the measured birefringence data are not affected. However, the endothelial cell density (ECD) for these corneas was less than the clinical transplantation grade cut-off, which is 2500 cells/mm$^2$. The Tenets of the Helsinki protocol were followed for handling the tissues. The study was approved by the institutional ethics committee (IEC, IEC Protocol No. IITH/IEC/2019/05/14, Dt. 2 May 2019), Indian Institute of Technology Hyderabad, India and institutional review board (IRB, Ethics. Ref. No. LEC 05–19–260, Dt. 14 May 2019), L V Prasad Eye Institute, Hyderabad, India. The inclusion criteria for the donor corneas is the suitability criteria, which is the endothelial cell density of less than 2500 cells/mm2 and optimum optical quality. There was no lower limit for the endothelial cell density. However, the optical quality of the corneas was assessed for the presence of excessive number of stromal folds and tissue swelling. Based on these inspections, the corneas were included in the study. The central distance between isotropic points (IPs; zones of zero birefringence, refer to our previous work [32]) (4–6 mm) was also chosen as inclusion criteria to ensure the similarity in birefringence across all the tested corneas. The twelve corneas were narrowed down to Five based on the inclusion criteria. The individual cornea was given a nomenclature that includes the sequence number C1 to C5, see Table 1.

**Table 1. The test matrix used in the present study to investigate the effect of sutures and host-donor alignment on corneal birefringence.**

| Host-Donor Misalignment/Sutures | No. of Cornea (ID) |
|---|---|
| 0˚ | 1 (C1) |
| 30˚ | 1 (C2) |
| 45˚ | 1 (C3) |
| 60˚ | 1 (C4) |
| 90˚ | 1 (C5) |

The birefringence data were acquired sequentially, as per **Table 1**, for all the corneas at the IOP of 20 mm Hg. A typical illustration of the *in-vitro* model of PK elucidating the structural misalignment of the host and the donor cornea is shown in **Fig 2**. The schematic of suture placement is shown in **Fig 2B**. Note that each cornea was imaged separately to obtain control birefringence data using the methodology described in Ref. [32]. Here, the control corneas are referred to as the corneas 'Before PK (B-PK)'. The control birefringence data was obtained 2 days before the PK procedure. After obtaining the control data, a small cut was placed on the scleral part of each corneoscleral button to mark its orientation with respect to the test cell/cornea mount (see **Fig 2C**). This helped in identifying the reference position of the button for creating known misalignment and birefringence measurement post-suturing. Then, the B-PK corneas were stored in MK medium at 4˚C till the PK was performed.

For performing PK, the corneas (C1 to C5) were mounted on a stainless steel artificial anterior chamber (LVPEI make), and a central 7–8 mm of the cornea was trephined (mechanical punched-out) using a stainless steel (SS) trephine. The diameter of the SS trephine was chosen to be in the range of 7–8 mm, depending upon the diameter of the cornea, as the sutures may enter the limbus if a larger trephine is used for the small size cornea. The trephined graft was then rotated by 0˚, 30˚, 45˚, 60˚ and 90˚ using Gimble/Mendez ring for the corneas C1, C2, C3, C4, and C5, respectively, to create a structural misalignment and then reattached to the same button using 16 interrupted 10–0 non-absorbable nylon sutures, followed by birefringence imaging at the IOP of 20 mm Hg. It is important to note that the birefringence data of the A-PK corneas (corneas after performing PK, denoted as A-PK) were obtained 2 days after performing the PK procedure. Previous studies have shown that cornea maintains its endothelial viability and ultrastructural integrity up to 90 hours in the MK medium [37, 38]. Therefore, the birefringence data of the control data was taken in the same day of procurement from eye bank and followed by PK procedure within two days. Before imaging, the visual inspection of the PK corneas showed no signs of deterioration in their optical quality. Although imaged at different timepoints, it was ensured that all the corneas (control and PK) were imaged in the window of 90 Hours after the procurement to prevent the influence of tissue deterioration on the measurements. Note that the donor cornea in the present study refers to the central part of the cornea, which was trephined from the Corneoscleral button and then reattached to the same button.

It is important to note that the length of the sutures on either side of the host-donor junction was kept to a maximum of 1 mm for creating a compression zone between the consecutive sutures to ensure a leak-proof joint. The sutures were placed up to 90–95% of the corneal thickness to ensure the absence of an internal wound gap, which may arise from inadequate suturing depth [15]. For data comparison, the suture near to 12'O clock position (measured clockwise with respect to scleral cut, see **Fig 2B** and **2C**) was marked as suture 1 (S1), and the subsequent sutures were marked sequentially in a clockwise manner (see **Fig 2C**). During

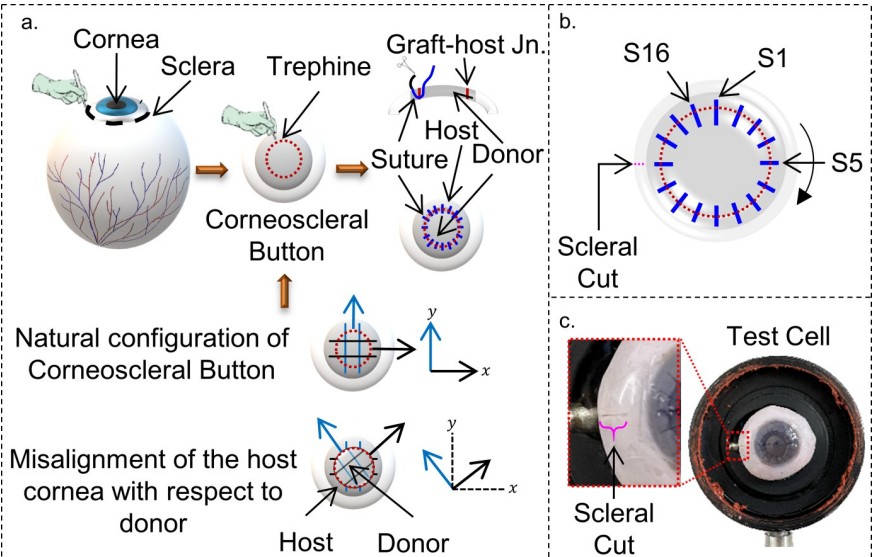

**Fig 2. An illustration of the in-vitro PK model used in the present study.** (A) The central portion of the corneoscleral button is trephined and rotated from its natural configuration to create structural misalignment, followed by suturing. (B) The sutures were placed clockwise, starting from the first suture (S1) at a 12'O clock position. (C) A small cut on the sclera (highlighted as a magenta colour dotted line) was made as a reference for creating misalignment and birefringence measurement in A-PK corneas.

suturing, the corneal surface was intermittently irrigated with saline solution to maintain the corneal hydration.

Intraoperative assessment of suture tension is crucial for even tissue distribution after keratoplasty. Although no objective measure currently exists for measuring the suture tension, this study utilized standard clinical methods to maintain optimal tension, preventing complications like humor leakage, tissue dimpling around the suture bite, etc. To elucidate further, the suturing process involves holding the donated cornea with delicate, double-pronged forceps at the point where the epithelial and stromal layers meet. The suture is then carefully threaded beneath the teeth of the forceps, passing through both the donated and the recipient tissues, ensuring that the suture penetrates to a depth of roughly 90% to minimize the risk of wound separation. The suture is secured by initially creating a triple loop, followed by two individual loops, or by using a slipknot that enables precise adjustment of tension. Additionally, the sutures in the present study were placed by an experienced corneal surgeon to avoid inter-surgeon variability in suture tension assessment.

## Data extraction

A binary mask with an approximately 1 mm region of interest (ROI) around all 16 sutures was created in MATLAB, and the maximum value of birefringence (fringe order) was obtained within each ROI. The maximum birefringence was obtained for all the B-PK and A-PK corneas. A typical representation of the data extraction procedure is shown in **Fig 3E**. The maximum birefringence obtained around each suture bite for B-PK and A-PK corneas for various degress of misalignments is given in the supporting excel S2 File. Further, the full-field birefringence images of the corneas were analyzed using ImageJ software to evaluate the influence of misalignment and suturing on the locations of the IPs. It is important to note that the binary mask was obtained for each cornea separately using its image under white light illumination (for example, see **Fig 3A** for cornea C2). The evaluated binary mask was applied to the

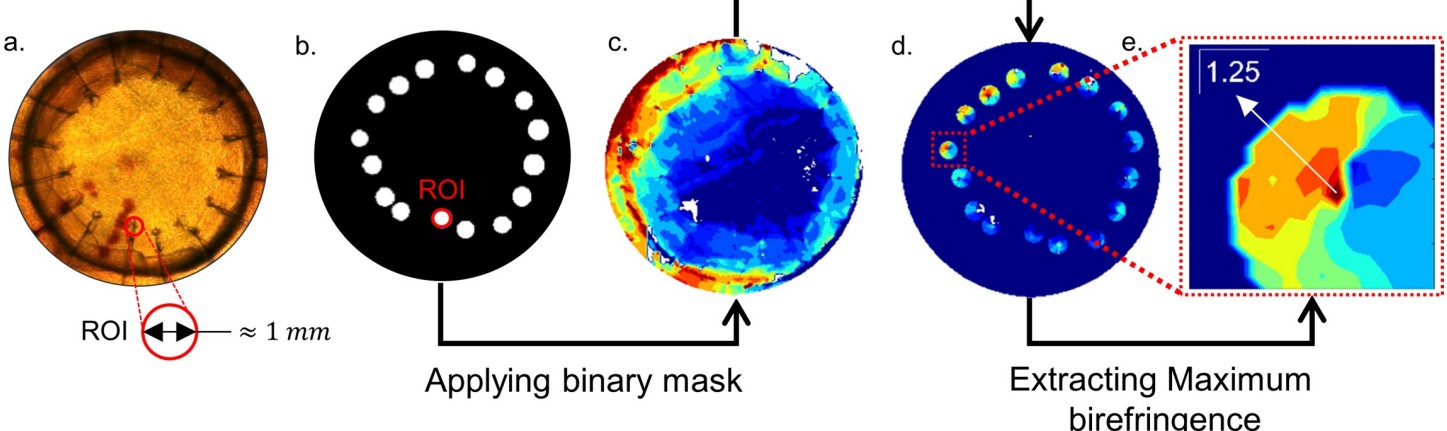

**Fig 3. Binary masking operation around the sutures and data extraction procedure.** (A) The Cornea C2 under white light illumination depicting the 1 mm region of interest (ROI) around the suture bite. (B) The binary mask to evaluate maximum retardation around each suture in B-PK and A-PK corneas. (C) The retardation map of the cornea C2 A-PK. (D) the retardation patterns around each suture bite obtained by masking the binary map (B) over (C). (E) the retardation distribution around the suture S13 depicting a lobe-like morphology.

birefringence map of the corresponding cornea before PK, and the maximum birefringence was extracted before PK at the location of each suture bite. This method facilitated the comparison of birefringence data before and after PK.

## Data analysis

In the present study, five groups of experiments were performed. In each group, N = 16 data points (corresponding to 16 simple interrupted sutures) were assessed before and after PK with different degrees of misalignments. The birefringence data obtained for B-PK and A-PK corneas were analysed in OriginPro (Version 9.0) using the methodologies presented in this section for understanding the qualitative and quantitative effects of the suturing and host-donor misalignment on corneal birefringence.

The birefringence data, N = 16 (for both B-PK and A-PK corneas), were checked for normality using the *Shapiro-Wilk* test. Then, One-way *ANOVA* with the *post-hoc* Tukey method was used to evaluate the birefringence differences around the suture bites' locations among all the B-PK corneas. Further, to understand the effect of suturing, the comparison between the B-PK and A-PK corneas for each case of misalignment was made using paired sample *t*- test with a significance level of 0.05. Later, the *Kruskal-Wallis* test was used to test statistical differences amongst all the A-PK corneas to understand the influence of misalignment. Additionally, the pair-wise comparisons of the A-PK corneas were made using the *Mann-Whitney* test with a significance level of 0.05.

## Results

Using the data analysis procedure described in section Introduction, the present section provides the statistical analysis of the birefringence in B-PK and A-PK corneas. It elucidates both qualitative and quantitative effects of the suturing and host-donor misalignment on corneal birefringence.

The PK procedure changes the local (around the suture bites) as well as the global morphology of the corneal isochromatics. For a better understanding of the local effects of suturing on the birefringence behaviour of the cornea, an example of the isochromatic distribution in the

cornea C2 around the suture bites is shown in **Fig 3C–3E**. It can be seen that the isochromatics around the suture bite exhibit a lobe-like distribution (see **Fig 3E**). This lobe-like morphology is analogous to the stress-concentration contours typically observed around a crack tip [35]. Additionally, the intensity (birefringence value) of these lobes seems to vary from suture to suture depending upon the applied suture tension (see **Fig 3D**). It can also be observed that incorporating the sutures changes the morphology of the peripheral isochromatics globally, as indicated by the alteration in the rhomboidal-type distribution of peripheral isochromatics typically observed in human corneas [32].

**Fig 4A** shows the box-whisker plot of the birefringence for the B-PK corneas, C1 to C5. It can be observed that the median (middle line within the box) and the mean (dot within the box) lies closer to each other for the corneas C1 to C3. Also, the birefringence is equally distributed around the median line for the corneas C1 to C3, indicating the normal distribution of the birefringence data. The *Shapiro-Wilk* test shows that the birefringence is normally distributed for all the B-PK corneas (See **Table 2**) at the significance level of 0.05. Interestingly, the birefringence distribution of cornea C4 and C5 shows a slight positive skew (mean>median) and negative skew (mean<median), respectively, depicting the asymmetric distribution of birefringence (See **Fig 4A**). After confirming the normality, the One Way *ANOVA* test was used for comparing the means of B-PK corneas, which showed that the mean birefringence (fringe order) was statistically the same for all the B-PK corneas (p = 0.264>0.05). Additionally, the *post-hoc Tukey* method was used for conducting the pairwise comparison of all the B-PK corneas. As shown in **Fig 4B**, the confidence interval band for each pair passes through the zero line. This confirms the One Way *ANOVA* results and shows that the birefringence distribution is statistically similar amongst all the B-PK corneas. **Table 2** also gives the normality test results for the A-PK corneas. It can be seen that, unlike the corneas C1, C3, and C5, the birefringence around the suture bites for corneas C2 and C4 follows a non-normal distribution.

**Fig 5A** shows the box-whisker plot for the A-PK corneas, C1 to C5. It can be seen that the birefringence distribution for the corneas C1, C3, and C5 shows approximately zero skew (mean ≈ median), indicating normal distribution, as shown by the normality test (see

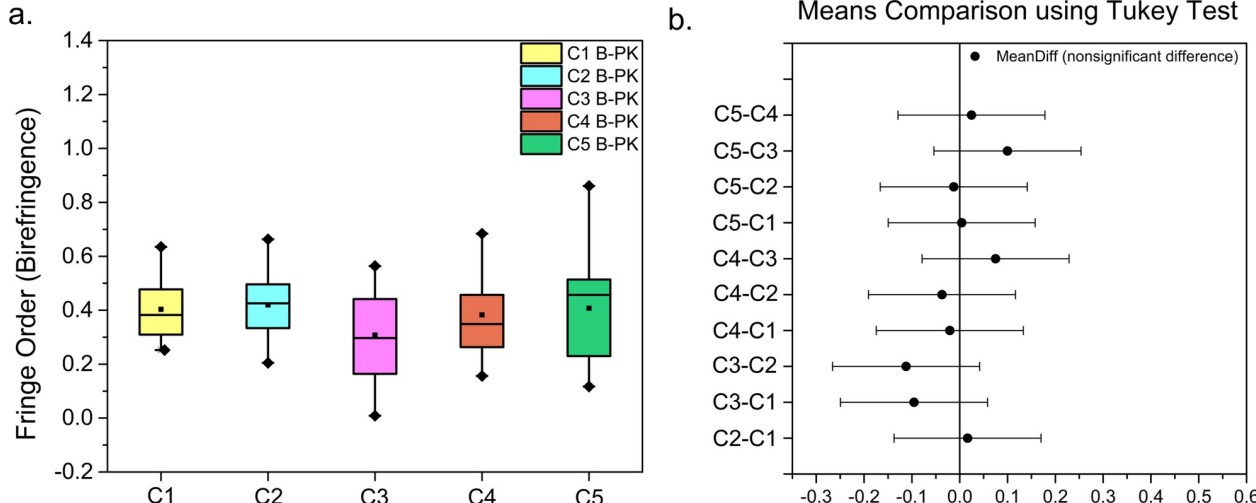

**Fig 4. Statistical Analysis results of B-PK corneas.** (A) The Box-Whisker plot of the distribution of the maximum birefringence around the locations of the suture bites for the corneas before PK (B-PK). (B) pair-wise comparison of the average of the maximum birefringence around the suture bites locations for B-PK corneas.

**Table 2. Normality test results for the birefringence around the locations of suture bites for the B-PK and A-PK corneas (Using the Shapiro-Wilk Test) at the significance level of 0.05.**

| Cornea | Category | Shapiro-Wilk Test (p-value) | Decision |
|--------|----------|------------------------------|----------|
| C1 |        | 0.250 | Normally distributed |
| C2 |        | 0.876 | Normally distributed |
| C3 | B-PK   | 0.683 | Normally distributed |
| C4 |        | 0.241 | Normally distributed |
| C5 |        | 0.932 | Normally distributed |
| C1 |        | 0.141 | Normally distributed |
| C2 |        | 0.024 | NOT Normally distributed |
| C3 | A-PK   | 0.096 | Normally distributed |
| C4 |        | 0.009 | NOT Normally distributed |
| C5 |        | 0.914 | Normally distributed |

Table 2). However, the corneas C2 and C4 show a positive skew in the birefringence distribution and indicate a non-normal distribution in birefringence (as depicted in Table 2). Interestingly, the birefringence distribution for the corneas C3 and C4 contains outliers, as shown in Fig 5A. These outliers represent comparatively tight sutures with respect to the other sutures within the respective cornea leading to non-uniform suturing. To analyse the effect of suturing, the birefringence of the B-PK and A-PK corneas was compared for each case of misalignment. Fig 5B shows the bar graph comparing the average of the maximum birefringence observed around all 16 ROIs (see Fig 3B) for each cornea before and after PK. The corresponding statistical comparison at the significance level of 0.05 is given in Table 3. As expected, the mean birefringence significantly increases due to the suturing process across all the misalignments cases except for the 60° misalignment case (see Table 3 and Fig 5B).

Based on the presence of outliers in the birefringence distribution of the A-PK corneas (C3 and C4), the *Kruskal-Wallis* test was utilized to compare the median birefringence of all the A-PK corneas at a significance level of 0.05 to test the hypothesis proposed in 'Introduction'. It is interesting to note that no significant difference in birefringence was found among all the A-PK corneas ($\chi^2 = 8.423$, p = 0.07). A pair-wise comparison was also performed on the birefringence distribution of the A-PK corneas using the *Mann-Whitney* test, as shown in Table 4. It can be seen that no statistical difference was observed between the all the pairwise misalignment cases. Therefore, for the all above A-PK pairs, it is difficult to reject the null hypothesis that the misalignment will not change the local birefringence measured around the suture bites.

From the above analysis, it can be understood that the PK procedure locally increases the birefringence around the suture bites. However, it cannot be disregarded that the misalignment is playing an observable role in two ways: (i) by exhibiting contrasting results on the statistical differences in the local birefringence observed around the suture bites for the A-PK pairs, as shown in Table 4 (ii) by creating the birefringence distribution in the ROIs' of A-PK corneas, as shown in Fig 5A. Therefore, to obtain more insights into the role of host-donor misalignment in PK, the global behaviour of the birefringence for the B-PK and A-PK corneas are investigated below.

Apart from the local changes in the corneal birefringence due to PK, as discussed earlier, it also introduces global changes in the birefringence behaviour of the cornea. Fig 6 shows the full-field phase maps (isoclinics and isochromatics) of the B-PK and A-PK corneas. Qualitative observation of the isochromatics of the B-PK and A-PK corneas reveals that the typical rhomboidal-type morphology of the peripheral isochromatics of the B-PK corneas significantly changed in the A-PK corneas. It is also evident in the corresponding isoclinic phase maps of

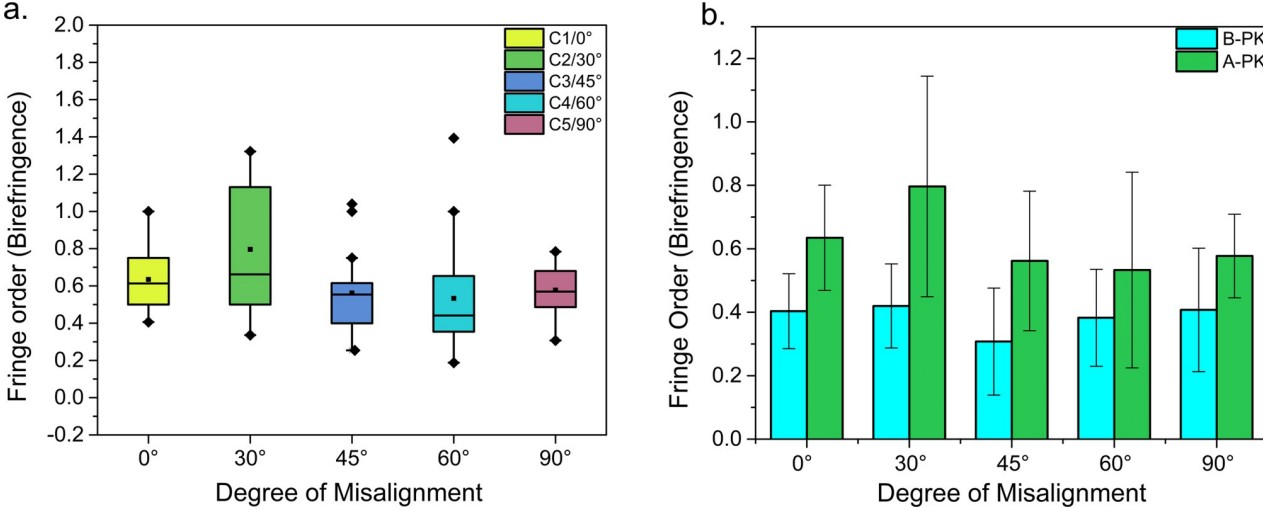

**Fig 5. Statistical Analysis results of A-PK corneas and their comparison with B-PK corneas.** (A) The Box-Whisker plot of the maximum birefringence distribution around the suture bites for each case of misalignment. (B) The comparison of the average of the maximum birefringence observed around all the suture bites for each cornea before and after PK. Note that B-PK denotes before PK, and A-PK denotes after PK.

the B-PK and A-PK corneas. For more details on the relationship between corneal isoclinics and isochromatics, the readers can refer to the previous work by the authors [32]. The location and the distance between IPs are the important global features of corneal birefringence [32]. Therefore, to quantify the global changes in the birefringence behaviour of the B-PK corneas after PK, the change in the disposition of the IPs was characterized by calculating the angular shift (observed relative misalignment) of the IPs and the distance between them. **Fig 6** shows the angular shift ($\alpha$ in degrees) of the IPs for all the A-PK corneas. The A-PK cornea C1 with 0˚ misalignment exhibited no angular shift of its IPs (see **Fig 6**). However, the A-PK corneas (C2 to C5) showed peculiar behaviour with respect to the angular shift of the IPs.

 **Fig 7** gives the bar graph comparing the distance between the IPs for the B-PK and A-PK corneas. It is intuitive that mechanical trephination relaxes the donor corneal tissue and can reduce the distance between the IPs. After performing PK, the distance between IPs should slightly increase due to the stretching of the tissue caused by suture tension and IOP. As expected, the B-PK cornea C1, C3, C4, and C5 exhibited a decrease in the distance between their respective IPs after PK (see **Fig 7**). In contrast, cornea C2 exhibited a 0.4% (marginal) increase in the distance between its IPs after PK (see **Fig 7**), while exhibiting an opposite angular shift (clockwise, -64˚) in the orientation of its IPs with respect to the induced anticlockwise misalignment of 30˚ (see **Fig 6**). It is interesting to observe that, unlike the C2 cornea, the corneas with 45˚, 60˚ and 90˚ (C3 to C5) misalignments underwent an angular shift of 25˚, 83˚

**Table 3. Effect of suturing on the maximum birefringence observed around the suture bites.** Note that B-PK denotes before PK, and A-PK denotes after PK.

| Cornea ID/Degree of Misalignment | Birefringence (Mean ± SD) | | Increase in Birefringence (%) | p-value | Decision |
|---|---|---|---|---|---|
| | B-PK | A-PK | | | |
| C1/0˚ | 0.403 ± 0.118 | 0.634 ± 0.165 | 57.3 | 0.001 | Statistically different |
| C2/30˚ | 0.419 ± 0.132 | 0.796 ± 0.347 | 89.9 | 0.0019 | Statistically different |
| C3/45˚ | 0.307 ± 0.168 | 0.561 ± 0.220 | 82.7 | 0.0006 | Statistically different |
| C4/60˚ | 0.382 ± 0.152 | 0.533 ± 0.308 | 39.5 | 0.08 | NOT statistically different |
| C5/90˚ | 0.407 ± 0.194 | 0.577 ± 0.132 | 41.7 | 0.014 | Statistically different |

**Table 4. Pair-wise comparisons of birefringence between the A-PK corneas C1 to C5 at the significance level of 0.05.** Note that the following comparisons were performed to understand the role of host-donor misalignment on the increased birefringence (from suturing). The Bonferroni-corrected p-value was used to arrive at the significance conclusion.

| Group1 –Group 2 | Test Statistic, Z | p-value | Bonferroni-corrected p-vaue | Significance |
|---|---|---|---|---|
| 0˚ - 30˚ | -0.088 | 0.373 | 0.99 | NOT statistically different |
| 0˚ - 45˚ | 1.021 | 0.307 | 0.99 | NOT statistically different |
| 0˚ - 60˚ | 2.232 | 0.025 | 0.25 | NOT Sstatistically different |
| 0˚ - 90˚ | 0.775 | 0.443 | 0.99 | NOT statistically different |
| 30˚ - 45˚ | 1.489 | 0.136 | 0.99 | NOT statistically different |
| 30˚ - 60˚ | 2.395 | 0.016 | 0.16 | NOT Sstatistically different |
| 30˚ - 90˚ | 1.225 | 0.220 | 0.99 | NOT statistically different |
| 45˚ - 60˚ | 0.961 | 0.336 | 0.99 | NOT statistically different |
| 45˚ - 90˚ | -0.060 | 0.546 | 0.99 | NOT statistically different |
| 60˚ - 90˚ | -1.523 | 0.127 | 0.99 | NOT statistically different |

and 59˚ (anticlockwise) in the orientation of their IPs (see **Fig 6**) and exhibited a 34%, 30%, and 22% decrease in the distance between their IPs (see **Fig 7**), respectively.

Here, the greater spread of birefringence (towards the higher suture tension) for the A-PK cornea C2 (see box-whisker plot, **Fig 5**A) is responsible for the increased distance between the IPs observed after PK. Additionally, the induced 30˚ misalignment and higher suture tension contribute to a greater topographic irregularity as indicated by the -64˚ change in the orientation of IPs. Similarly, for the other A-PK corneas (C3 to C5), the decreased distance between IPs can be attributed to the lower suture tension as compared to the cornea C2 (see **Fig 5A**). Further, the angular shift of the IPs of the corneas C3 to C5 exhibits erratic behaviour depicting a complex interplay of the suture tension and host-donor misalignment (see **Fig 6**).

The results on the birefringence behaviour of the B-PK and A-PK corneas presented here reveal the synergistic effect of suture tension and host-donor misalignment to create topographic irregularities in PK. The suture-tissue interaction manifests itself in two ways: (i) a

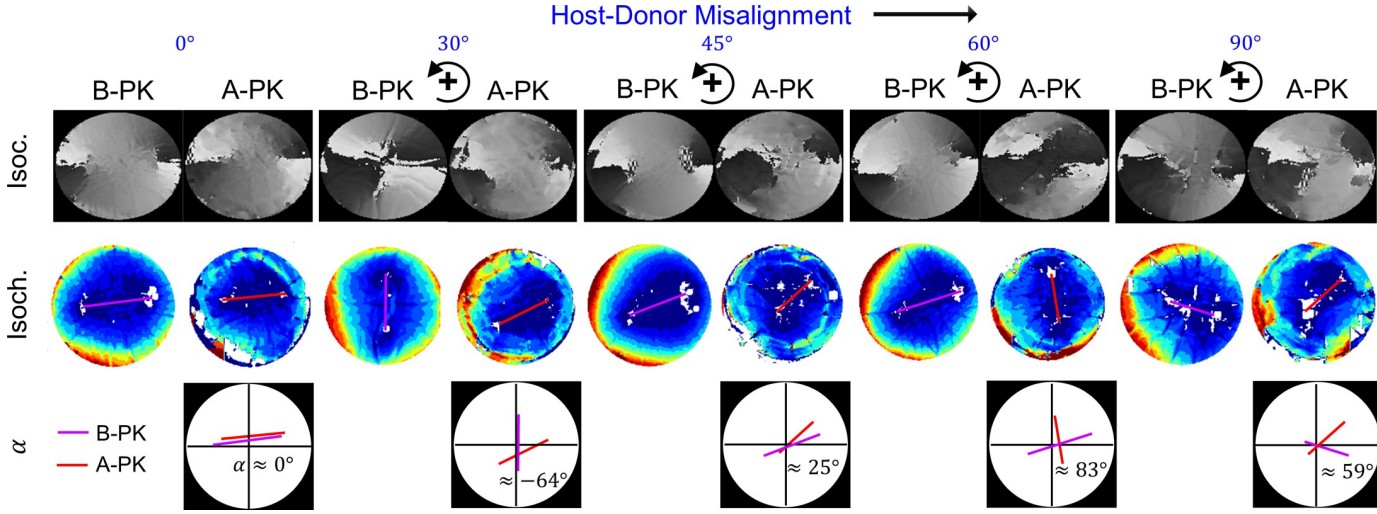

**Fig 6. Effect of suturing and misalignment on the angular displacement of isotropic points.** Note that B-PK denotes the cornea before PK, and A-PK denotes the cornea after PK. Isoc. denotes the Isoclinic phase map and Isoch. denotes isochromatic phase map, while $\alpha$ represents the angular shift between the line joining the isotropic points before and after PK. Again, it is important to note that the trephined buttons were rotated anticlockwise to realize the host-donor misalignments.

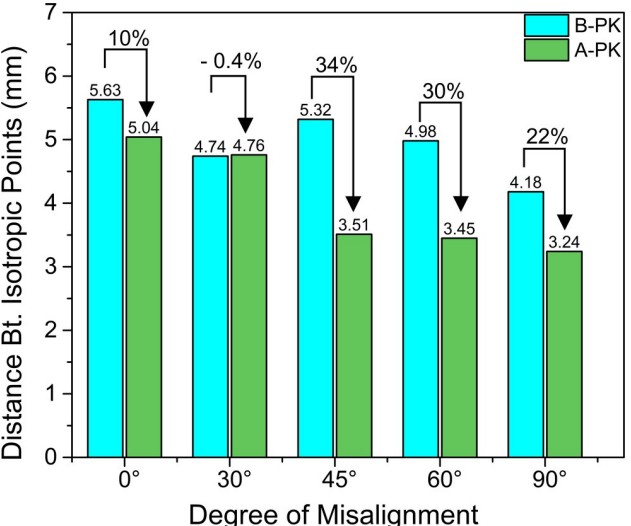

**Fig 7. Effect of suturing on the distance between isotropic points for each case of misalignment.** Note that B-PK denotes the cornea before PK, and A-PK denotes the cornea after PK.

local increase in birefringence around suture bites and (ii) a global change in the features of the corneal birefringence (erratic change in the angular orientation of IPs as shown in **Fig 6**).

## Discussion

In PK, the minimization of post-operative astigmatism is one of the major concerns for the cornea surgeon. Suture tension, suture depth, suture bite, suture length, and suture radiality are some of the factors that are considered crucial in determining the level of PPKA. However, host-donor structural misalignment is an unaddressed factor that plays a key role in the genesis of PPKA. Therefore, the goal of the present study was to highlight the role of the microstructural matching of the host and the donor cornea during corneal transplant surgery and its influence on topographic irregularities.

Incorporating sutures in the cornea creates a material discontinuity at the location of the suture bite, which acts as a stress-raiser characterized by the morphological change in the isochromatics around the suture bite (see **Fig 3C** and **3**D). Although local, this morphological variation around all the suture bites results in an overall global change in the morphology of the peripheral isochromatics. It is known that the collagen microstructure of the peripheral cornea is vital in stabilizing the corneal curvature [39]; therefore, adequate suture tension distribution is necessary to ensure it during PK. This condition holds when the donor tissue is structurally aligned with the host eye, as indicated by the near-rhomboidal morphology of peripheral isochromatics for the A-PK cornea C1 (see **Fig 6**). However, a standard operating procedure of PK does not comprise the microstructural matching between the host and the donor cornea. Therefore, alongside uneven distribution in suture tension, a host-donor structural misalignment induces additional irregularities in corneal curvature, as indicated by the distortion in the rhomboidal morphology of the peripheral isochromatics for the A-PK cornea C2 to C5 (See **Fig 6**).

Each suture bite acts as a stress-raiser, and incorporating sutures significantly increased the birefringence around the suture bites (see **Table 3**), except for the A-PK cornea C3 (with 60˚ host-donor misalignment). This might be due to insufficient suture tension applied during the suturing in C3. It is important to note that although the sutures significantly increased the

birefringence in each case of misalignment, the increased birefringence remains statistically the same amongst the misaligned pairs (see **Table 4**). Suffice it to say that the degree of the host-donor misalignment does not play a significant role in the local increase in corneal birefringence around the suture bites.

Interestingly, though misalignment does not influence the increase in the local birefringence around the suture bites, it significantly and peculiarly alters the disposition of the IPs over the corneal surface (see **Fig 6**). Ignoring the structural misalignment aspect, i.e., the induced misalignment in the B-PK corneas (C2 to C5), should have caused no angular shift of their IPs after PK (similar to A-PK cornea C1). In other words, the observed relative misalignment $\alpha$ should be zero for all the A-PK corneas. However, all the cases of misalignments (A-PK C2 to C5) have shown erratic angular shifts of their IPs. Multiple studies have shown the occurrence of topographic irregularities in the cornea after PK [25, 40]. The authors believe that the peculiar change in the disposition of the IPs over the corneal surface indirectly suggests topographic irregularity, as the disposition of the IPs depends on both curvature and stress distribution in the cornea. Based on the present findings, the topographic irregularity observed in PK corneas [41] seems to be a coupled phenomenon involving an interplay of suture tension and structural misalignment.

Typically, the mitigation strategy for such irregularities involves suture removal in multiple sittings [5, 41, 42]. It is known that post-PK, the host-donor wound characteristics [43] decide the corneal topography [4]. The wound characteristics depends on the distribution of the suture tension. Based on the birefringence measurement, the present study shows that the structural misalignment between the host and donor tissue in conjunction with suture tension leads to these topographic irregularities. Hence, the present study points towards the indirect effect of misalignment on wound characteristics, which can be managed intraoperatively by birefringence imaging. The misalignment is influencing the surgeon to put sutures in a way so that tissue distribution becomes non-uniform, which eventually affects the suture tension and wound apposition characteristics [4]. Therefore, birefringence imaging presents a viable means to manage these post-PK complications intraoperatively by structurally matching the host and donor tissue.

In the present study, the topographical irregularities are quantified in terms of the relative disposition of the IPs before and after PK. This quantification is straightforward in *in-vitro* scenarios due to the ease of generating control data (B-PK). However, the quantification of irregularities in terms of the relative disposition of IPs is challenging for *in-vivo* conditions. The inter-individual variability in corneal birefringence makes it challenging to determine the control data that represents a standard for comparison. In other words, quantifying the absolute disposition of IPs and their association with corneal astigmatism is required to realize a metric that dictates the influence of corneal microstructure on the optical outcome of PK. Therefore, multiple experimental studies with large human cornea samples are necessary to establish detailed insights on the implications of structural aspects of the cornea on astigmatism in both pre and post-surgical scenarios. Nevertheless, the results of the present study support the importance of structural matching between the host and the donor cornea during PK.

The authors suggest that once the host and the donor cornea are structurally matched, the sutures should be placed such that a consistent distribution of suture length, depth, radiality, and tension is ensured to compensate for the directional resistance offered by microstructural anisotropy. To facilitate the structural alignment of the host and the donor cornea during the PK procedure, an *in-vivo* device capable of real-time imaging of the birefringence of the cornea is required. The authors are dedicatedly working towards the realization of such a device, mainly based on the technique of digital reflection photoelasticity.

Nahum et al. [44] showed that the Host-Donor misalignment may not influence the optical outcome in Descemet Stripping Automated Endothelial Keratoplasty (DSEAK). The use of ultrathin grafts and the absence of suturing might be responsible for the insignificant contribution of donor's structural anisotropy on the overall mechanical response of the host eye after transplantation. In contrast, the present study reveals the significant contribution of host-donor misalignment on the topographical irregularities in the cornea undergoing PK.

The present study suffers some limitations. Firstly, the present study included only five corneas to test the proposed hypothesis. While it is true that the sample size is limited, it is crucial to understand that for this in-vitro model, we require transplantation-grade human tissues to validate, which are not readily available. The transplantable grade human donor corneas are scarce, particularly in developing countries with a long waiting list for transplantation. Further, the small sample size does not replicate the real-world situation as this study is performed in a controlled experimental setup where the authors tried to look or discern the impact of one factor (structural misalignment) on topographical irregularities. In a real-world scenario, it would be very difficult to understand the influence of only one factor on the overall topographical irregularities because there are so many other factors are involved. Secondly, since the same skilled surgeon performed the suturing in all the corneas, the authors assume similarity of the factors such as suture radiality and suture length on either side of the host-donor junction depth and distance of the sutures from the optical centre amongst all the corneas. Addtionally, the PK procedure was spread over two days, and then the birefringence data was taken for all the corneas which can potentially affect the measurements, due to changes in mechanical strength and integrity.

Another important limitation of the study is its small and focused nature, where the authors tried to understand the impact of one of the less explored factors (degree of structural misalignment) on topographical irregularities of the cornea, which is not taken into consideration currently while performing PK. While it is true that clinically there are other factors such as wound healing, patient characteristics and surgical technique, etc. that are involved, and one may cancel out the other. However, the findings of the present study highlight the impact of the misalignment and suture tension on topographical irregularities in the cornea and suggest the importance of structurally aligning the host and donor to control post-PK astigmatism.

Further, the present study considers two quantities, namely, angular change in the IPs and distance between IPs that represent the disposition of the IPs over the corneal topography after PK. However, a third factor that gives offset between the line joining the IPs also quantifies the disposition of the IPs. The present study disregards the third factor, as the other two quantities appear to be sufficient to investigate the coupled effect of suture tension and misalignment on the corneal birefringence.

In conclusion, local suture tension increases the birefringence around the suture bite. Interestingly, the structural misalignment between the host and the donor tissue does not greatly influence the locally increased birefringence due to sutures. However, the structural misalignment and suture tension collectively influence the suture-tissue interaction resulting in an erratic change in the disposition of IPs after PK. These erratic changes depict topographic irregularities in the corneas. Therefore, the zero-degree structural misalignment of the host and the donor tissue is preferable to minimize the topographical irregularities and related astigmatism post-PK.

Clinical Relevance: In the current clinical practice, the corneas harvested for transplantation are not marked based on their structural anisotropy. Therefore, the harvested donor corneas are trephined and randomly sutured to the host eye without paying any attention to their microstructural aspect (SI (superior-inferior)-NT (nasal-temporal) directions). In the present study, our interest was to investigate whether the host-donor misalignment affects corneal

birefringence (microstructural aspect). Although no reliable or significant effect of misalignment was observed on the overall birefringence around the sutures, the host-donor misalignment in penetrating Keratoplasty contributes to the topographical irregularities in the cornea as indicated by the erratic change in the disposition of isotropic points. Based on the local and global behaviour of the corneal birefringence in A-PK corneas, the authors propose that it may become more difficult for the surgeon to maintain better suture tension in misaligned corneas. It is because the misalignment induces unforced surgeon error in putting sutures that hold the regular topography of the cornea. Based on our study, it is recommended that incorporating an additional step of aligning the host and the donor cornea in a typical surgical process may improve the refractive outcome of PK.

## Supporting information

**S1 File. The birefringence images of all the B-PK and A-PK corneas for all the degress of misalignment.**
(ZIP)

**S2 File. The maximum birefringence extracted around the 16 suture bites for B-PK and A-PK corneas for various degrees of misalignment.**
(XLSX)

## Author Contributions

**Conceptualization:** Himanshu Gururani.

**Data curation:** Himanshu Gururani.

**Formal analysis:** Himanshu Gururani, Viswanath Chinthapenta, Sayan Basu.

**Funding acquisition:** Viswanath Chinthapenta.

**Investigation:** Himanshu Gururani, Minal Doulatramani.

**Methodology:** Himanshu Gururani, Sai Naga Sri Harsha Chittajallu, Ramji M.

**Project administration:** Viswanath Chinthapenta, Sayan Basu, Ramji M.

**Resources:** Sai Naga Sri Harsha Chittajallu, Viswanath Chinthapenta, Sayan Basu, Ramji M.

**Software:** Himanshu Gururani, Ramji M.

**Supervision:** Viswanath Chinthapenta, Sayan Basu.

**Validation:** Himanshu Gururani, Sayan Basu.

**Visualization:** Himanshu Gururani, Viswanath Chinthapenta, Sayan Basu.

**Writing – original draft:** Himanshu Gururani.

**Writing – review & editing:** Himanshu Gururani, Sai Naga Sri Harsha Chittajallu, Minal Doulatramani, Viswanath Chinthapenta, Sayan Basu, Ramji M.

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
