## [Decision Letter · Decision Letter 0]

16 Jun 2024

PONE-D-24-09388Collagen imaging reveals synergistic effects of sutures and host-donor misalignment on topographical irregularities in penetrating keratoplastyPLOS ONE

Dear Dr. M,

Thank you for submitting your manuscript to PLOS ONE. After careful consideration, we feel that it has merit but does not fully meet PLOS ONE’s publication criteria as it currently stands. Therefore, we invite you to submit a revised version of the manuscript that addresses the points raised during the review process.

We look forward to receiving your revised manuscript.

Kind regards,

Bhavana Sharma, MS

Academic Editor

PLOS ONE

Journal Requirements:

 [This study was funded by the intramural funds

of LV Prasad Eye Institute (LVPEI), Hyderabad; and partially sup-

ported under Prime Minister Doctoral Research Fellowship grant (to

author HG) by the Department of Science and Technology, India, Sci-

ence, and Engineering Research Board, India, Confederation of Indian

Industries, India, the Government of India & LVPEI.].  

Additional Editor Comments:

Please include the following as a limitation of study :

the PK procedure was

spread over two days, and then the birefringence data was taken for all the

corneas which can potentially affect the measurements , due to changes in mechanical strength and integrity .

Further please address the concerns of referees .

Reviewers' comments:

Reviewer's Responses to Questions

**Comments to the Author**

1. Is the manuscript technically sound, and do the data support the conclusions?

Reviewer #1: Yes

Reviewer #2: Yes

Reviewer #3: Yes

2. Has the statistical analysis been performed appropriately and rigorously? 

Reviewer #1: I Don't Know

Reviewer #2: I Don't Know

Reviewer #3: Yes

3. Have the authors made all data underlying the findings in their manuscript fully available?

Reviewer #1: Yes

Reviewer #2: Yes

Reviewer #3: Yes

4. Is the manuscript presented in an intelligible fashion and written in standard English?

Reviewer #1: Yes

Reviewer #2: Yes

Reviewer #3: Yes

5. Review Comments to the Author

Reviewer #1: The study is well conceived, performed rigorously and the manuscript describes it lucidly.

The introduction mentions DALK and FLAK as techniques that potentially minimize astigmatism compared to PK. This is not supported by evidence, and may be omitted.

Topography guided selective suture removal is generally a post operative procedure, and may be removed from intraoperative measures to reduce astigmatism.

Reviewer #2: Changes suggestions after Line 66:

Original:

[13,14]. Further, the techniques such as SSR, sequential suture removal [15,16], intra-stromal ring (ICRS) insertion [17], intraocular lens (IOLs) [18,19], Photorefractive Keratectomy [20,21],laser in-situ keratomileusis (LASIK) [22,23] and repeat keratoplasty [24] are used post-operatively to manage PPKA.

Suggestion:

[13,14]. Further, the techniques such as sequential suture removal (SSR) [15,16], insertion of intra corneal ring segments (ICRS) [17], intraocular lens (IOLs) [18,19], Photorefractive Keratectomy (PRK) [20,21],laser in-situ keratomileusis (LASIK) [22,23] and repeat keratoplasty [24] are used post-operatively to manage PPKA.

Reviewer #3: Query 1: The authors have selected five corneas for the experiment. Kindly provide more details about the criteria for tissue selection. It is mentioned that corneas with an endothelial cell count (ECC) of less than 2500 were selected. Were there any additional criteria for selection, such as a lower limit for ECC, corneal thickness, or the presence of stromal folds? Additionally, were any tests performed to assess the biomechanical properties of the selected corneal tissues?

Query 2: The absence of topographical data for the tissues makes it challenging to determine whether the change in orientation of the graft had any effect on the final birefringence pattern or apposition.

6. PLOS authors have the option to publish the peer review history of their article (what does this mean?). If published, this will include your full peer review and any attached files.

Reviewer #1: No

Reviewer #2: **Yes: **Frederico Bicalho Dias da Silva

Reviewer #3: **Yes: **Sonam Yangzes

---

## [Author Response · Author response to Decision Letter 0]

1 Jul 2024

Response to Editor

(Reviewer comments are shown in black-colored text, and responses to reviewer

comments are shown in blue-colored text)

Editor: (1) Please include the following as a limitation of study: the PK procedure was spread over two days, and then the birefringence data was taken for all the corneas which can potentially affect the measurements, due to changes in mechanical strength and integrity.

i. As suggested by the editor, the authors have incorporated the changes in the revised manuscript.

ii. The changes are reflected in lines 454-456, p21.

Response to Reviewer-1

(Reviewer comments are shown in black-colored text, and responses to reviewer

comments are shown in blue-colored text)

Reviewer #1: The study is well conceived, performed rigorously and the manuscript describes it lucidly.

The introduction mentions DALK and FLAK as techniques that potentially minimize astigmatism compared to PK. This is not supported by evidence, and may be omitted.

Topography guided selective suture removal is generally a post operative procedure, and may be removed from intraoperative measures to reduce astigmatism.

i. The authors thank the reviewer for the appreciating the rigor that was put into the manuscript and the work conducted by the authors.

ii. Based on the reviewer’s suggestions, the authors have omitted the DALK and the FLAK from the introduction section. 

iii. The authors agree with the reviewer that topography guided selective suture removal is used post operatively and therefore has been removed from the intraoperative measures. 

iv. As per the reviewer’s suggestion, the above points have been incorporated in the revised manuscript, lines 59-67, see p 3. 

Response to Reviewer-2

(Reviewer comments are shown in black-colored text, and responses to reviewer

comments are shown in blue-colored text)

Reviewer #2: Changes suggestions after Line 66:

Original:

[13,14]. Further, the techniques such as SSR, sequential suture removal [15,16], intra-stromal ring (ICRS) insertion [17], intraocular lens (IOLs) [18,19], Photorefractive Keratectomy [20,21],laser in-situ keratomileusis (LASIK) [22,23] and repeat keratoplasty [24] are used post-operatively to manage PPKA.

Suggestion:

[13,14]. Further, the techniques such as sequential suture removal (SSR) [15,16], insertion of intra corneal ring segments (ICRS) [17], intraocular lens (IOLs) [18,19], Photorefractive Keratectomy (PRK) [20,21],laser in-situ keratomileusis (LASIK) [22,23] and repeat keratoplasty [24] are used post-operatively to manage PPKA.

i. As per the reviewer’s suggestion, the above changes have been incorporated in the revised manuscript, lines 67-70, see p 3-4. 

Response to Reviewer-3

(Reviewer comments are shown in black-colored text, and responses to reviewer

comments are shown in blue-colored text)

Reviewer #3: Query 1: The authors have selected five corneas for the experiment. Kindly provide more details about the criteria for tissue selection. It is mentioned that corneas with an endothelial cell count (ECC) of less than 2500 were selected. Were there any additional criteria for selection, such as a lower limit for ECC, corneal thickness, or the presence of stromal folds? Additionally, were any tests performed to assess the biomechanical properties of the selected corneal tissues?

i. The inclusion criterion for the corneas were endothelial cell density of less than 2500. There was no lower limit for the cell density. However, the optical quality of the corneas was assessed for the presence of excessive number of stromal folds and tissue swelling. Based on these inspections, the corneas were included in the study.

ii. The authors would like to clarify that no additional tests were performed to assess the biomechanical properties of the cornea. However, as mentioned in the manuscript (see p7, line 149-154), the central distance between isotropic points (IPs; zones of zero birefringence, refer to our previous work (4-6 mm) was also chosen as inclusion criteria to ensure the similarity in birefringence across all the tested corneas.

iii. The authors have incorporated the details related to inclusion criterion in the revised manuscript, see p7, lines 150-153. 

Reviewer #3: Query 2: The absence of topographical data for the tissues makes it challenging to determine whether the change in orientation of the graft had any effect on the final birefringence pattern or apposition.

i. The authors agree with the reviewer that the absence of the topographic data makes it challenging to ascertain that the change in the graft’s orientation affects the final birefringence pattern. It is typically expected that the reorientation of the graft would topographically alter its position, and a 30-degree misalignment would result in a 30-degree change in the angular position of isotropic points on the corneal surface.

ii. However, the authors would like to emphasize that the peculiar change in the disposition of the IPs (Ex.: -64-degree change for 30-degree mismatch) over the corneal surface indirectly suggests the influence of structural mismatch between the graft and the host tissue on the overall topographic irregularity, as the disposition of the IPs depends on both curvature and stress distribution in the cornea. 

iii. Based on the present findings, the authors suggest that the topographic irregularity observed in PK corneas seems to be a coupled phenomenon involving an interplay of suture tension and structural misalignment between the graft and the donor.

iv. The authors believe that multiple experimental studies with large human cornea samples are necessary to establish detailed insights on the implications of structural aspects of the cornea on astigmatism in both pre- and post-surgical scenarios. Nevertheless, the results of the present study support the importance of structural matching between the host and the donor cornea during PK.

v. Additionally, structural marking could be introduced as an additional step in eye banking protocols, which would involve marking the harvested cornea in nasal-temporal and superior-inferior directions to facilitate the structural matching of the graft with the donor's eye by the surgeons.

---

## [Editor Report · Decision Letter 1]

19 Jul 2024

Collagen imaging reveals synergistic effects of sutures and host-donor misalignment on topographical irregularities in penetrating keratoplasty

PONE-D-24-09388R1

Dear Dr. Ramji M

We’re pleased to inform you that your manuscript has been judged scientifically suitable for publication and will be formally accepted for publication once it meets all outstanding technical requirements.

Kind regards,

Bhavana Sharma, MS

Academic Editor

PLOS ONE
---

## [Editor Report · Acceptance letter]

29 Jul 2024

PONE-D-24-09388R1 

PLOS ONE

Dear Dr. M., 

I'm pleased to inform you that your manuscript has been deemed suitable for publication in PLOS ONE. Congratulations! Your manuscript is now being handed over to our production team.

Kind regards, 

on behalf of

Dr. Bhavana Sharma 

Academic Editor

PLOS ONE